# Examining the Structure of Difficulties in Emotion Regulation Scale with Chinese Population: A Bifactor Approach

**DOI:** 10.3390/ijerph18084208

**Published:** 2021-04-15

**Authors:** Lingling Xu, Jialing Li, Li Yin, Ruyi Jin, Qi Xue, Qianyi Liang, Minqiang Zhang

**Affiliations:** 1School of Psychology, South China Normal University, Guangzhou 510631, China; xllpsy@m.scnu.edu.cn (L.X.); carrieli@m.scnu.edu.cn (J.L.); lydia@m.scnu.edu.cn (L.Y.); ruyi@m.scnu.edu.cn (R.J.); xq_psy@m.scnu.edu.cn (Q.X.); channyleung@m.scnu.edu.cn (Q.L.); 2Key Laboratory of Brain, Cognition and Education Sciences (South China Normal University), Ministry of Education, South China Normal University, Guangzhou 510631, China; 3Center for Studies of Psychological Application, South China Normal University, Guangzhou 510631, China; 4Guangdong Key Laboratory of Mental Health and Cognitive Science, South China Normal University, Guangzhou 510631, China

**Keywords:** emotion regulation, Difficulties in Emotion Regulation Scale, factor analysis, bifactor model

## Abstract

The Difficulties in Emotion Regulation Scale (DERS), as one of the most frequently employed measures of emotion regulation (ER), has increasingly been used in numerous researches and applications. However, the structures derived from previous factor-analytic studies have a high degree of inconsistency. In the current study, both the traditional factor analysis method and novel (bifactor) modeling approaches were employed to examine the most optimal measurement structure of the DERS in a sample of 1036 Chinese participants. After a series of comparisons, the findings indicated that the bifactor model, with a general ER factor and four distinct subdimensions, was the most optimal structure for the DERS. Based on the study’s findings, the discussion was focused mainly on the future directions and the implications of this bifactor model. The impact and limitations of the study were also discussed, and several suggestions for future research were provided at the end of the paper.

## 1. Introduction

Emotion regulation (ER) has been getting substantial and increased attention in the last several decades in psychology and related fields [1]. As a rough metric of research activities in this area, PsycINFO search, with “emotion regulation” as the keyword, yielded more than 219,719 search results in December 2020. The second edition of Handbook of Emotion Regulation [2] introduces an extensive body of research that has investigated the underpinnings of ER in many aspects, as well as its associations with physical and psychological health and dysfunction. Theoretical models associate successful ER with good health outcomes, improved relationships, and excellent academic and work performance [3]. Conversely, within the field of clinical psychology, poor ER is a transdiagnostic risk factor that has been connected to numerous psychological disorders, including anxiety, substance use, and eating disorders [4]. Considering the disorders listed above, understanding the nature and risk factors of ER has extremely important clinical implications.

To assess the degree of an individual’s ER, an increasing number of questionnaires were developed, such as the Negative Mood Regulation (NMR) [5], the Cognitive Emotion Regulation Questionnaire (CERQ) [6], the Emotion Regulation Questionnaire (ERQ) [7], the Difficulties in Emotion Regulation Scale (DERS) [8] and the Regulatory Emotional Self-Efficacy Scale (RESE) [9]. The DERS has become one of the most broadly employed screening instruments, as it can measure ER from a multidimensional profile [8,10]. In recent decades, the DERS has been employed in a variety of different circles, such as clinical psychology [11], psychological health [12], etc. Furthermore, several studies offered preliminary evidence for the high reliability and validity of the DERS [8,13]. It has therefore been translated into various languages, including English [8], Mexican [14], Spanish [15], and Chinese [16].

However, all of these studies were implemented under the assumption that the DERS has good conceptual and psychometrical assessment validity. Although many studies have investigated the factor structure of the DERS, it has been previously mentioned that ER is complex and may have different meanings for different people [17]. People with different cultures, experiences and social statuses, may have distinct strategies toward ER. Thus, the original factor structure of the DERS might not be valid for Chinese population. In addition, even in the same cultural environment, there may be subtle differences between different groups. To make the screening scales more effective and to guarantee comparability of measured properties across different groups of respondents, good psychometric invariance and clear factor structure are necessary.

### 1.1. Factor Structure of the DERS

The original DERS was developed by Gratz and Roemer [8] after they integrated key features of ER into a holistic concept. The factor analysis of the DERS items revealed six factors (non-acceptance, goals, impulse, awareness, strategies, and clarity), and this model can explain 55.68% of the total variance of variables (labeled the structure as “A”). In order to investigate the structure of DERS, some studies were carried out [10,13,14]. However, the structures were highly inconsistent and they were mainly focused on 4-, 5-, and 6-factor structures.

Marin Tejeda et al. [14] translated the DERS into Spanish version in a sample of 455 Mexican adolescents. The results of confirmatory factorial analysis (CFA) showed that the data fitted 4-factor model with 24 items (labeled as “B”). These four factors (non-acceptance, goals, awareness, and clarity) accounted for 45.3% of the total variance and showed good psychometric properties with high internal consistency and adequate concurrent validity. Compared with model A, the “strategies” factor was removed and the “impulse” factor was integrated into different factors in model B.

Bardeen et al. [10] conducted a series of CFAs in a sample of 1045 female undergraduate students from a mid-sized Midwestern University, resulting in a 5-factor structure. The results showed that the factor named “awareness” tended to share small inter-correlations with the other DERS factors and the revised 5-factor model (labeled as “C”) provided an adequate fit to the data after removing the “awareness” dimension. Model C had a good internal consistency and did not substantially sacrifice the standard correlation validity of the original scale. Another 5-factor structure was revealed in the sample of Chilean population [13], which accounted for 61.28% of the total variance (labeled as “D”). Model D with 5 factors (emotional exclusion, emotional out of control, emotional interference, emotional loosening, and emotional confusion) was ultimately derived from model A, but it was very different from model A.

For a 6-factor structure, Gómez-Simón et al. [15] conducted the first study using DERS to evaluate ER in a sample of Spanish adolescents. The study ended up with six highly explicable factors (non-acceptance, goals, impulse, awareness, strategies, and clarity) and the internal consistency for the subscales was moderate to satisfactory (labeled as “E”). Another similar 6-factor structure was found by Li, Han, Gao, Sun, and Ahemaitijiang [16] when they developed a Chinese version of DERS (labeled as “F”). The factor structure of model F was the same as that of model A, but four items in model A were deleted due to their weak factor loadings. The internal consistency, concurrent validity, and convergent validity of the scale were good. The test–retest reliability was also good but slightly lower than the original scale.

Potential reasons for the inconsistency among these studies may partly lie on theoretical differences, since the factor structure itself has not been uniformly defined in different studies. Discovering common factor structure is on the basis of consistent definition, which determines the domain of the construct and the item pool [18]. Whereas there seem to be additional methodological issues that can cause some degree of inconsistency across studies. One issue, which can be considered, is the sample size influencing factorial solutions. In the existing studies, sample size between 199 [19] to 1568 [20] has been employed. Besides, the factorial complexity of ER measures can be attributed to the method used to extract the number of factors when implementing exploratory factor analyses (EFA). For example, studies using the Kaiser-Guttman criterion [8] yielded more factors than studies employing maximum likelihood method [21]. Consensus on an optimal structure for ER measures is extremely important for ER studies [22], and achieving a consensus definition is a critical step before its common factor structure can be determined. Though there have been several attempts at theory building, there is also a lack of commonly used construct definition or theoretical view [23]. Moreover, because validation of the factor structure of a measure is a process driven by theory and empirical data, methodologically rigorous methods will inform our effort toward a consensus view of ER.

### 1.2. Bifactor Model

The bifactor model was originally developed by Holzinger and Swineford [24], and was regarded as an alternative model to non-hierarchical multidimensional models. An example of bifactor model with three special factors was showed in Figure 1.

The details of the model are listed as follows. If a scale is conducted with *p* items, the score of each item is presented as *X*_1_, *X*_2_,…,*X_p_* and this scale has measured a general factor *G* and *n* specific factors *S_1_,S_2_…,S_n_* then the observed variable *X_i_* can be presented as:(1)$$Xi=\alpha iG+\sum_{j=1}^{n}bijSj+\delta i,i=1,2,\ldots ,p$$
where *α_i_* is the loading of *X_i_* on general factor *G*, *b_ij_* is the loading of *X_i_* on special factor *S_j_*, and *δ_i_* is the measurement error of *X_i_* Furthermore, to make the model easier to converge and explain, it is usually assumed that the relationship between the general factor, specific factors and measurement error, are orthogonal.

Unlike the correlated-factors model utilized in previous studies, bifactor model hypothesizes that (a) there is a general factor that accounts for the commonality shared by the facets, and (b) there are multiple specific factors, each of which accounts for the unique influence of the specific component over and above the general factor [25]. Moreover, any remaining systematic covariation among the items would be captured by their loadings on the narrower specific factors. Therefore, the bifactor approach also makes it possible to estimate the relative sizes of the general and specific symptom components, and compare their independent contributions to the prediction of ex-ternal criteria. Through this approach, whether the degree of multidimensionality in a given measure is sufficient to support using subscales or not can be evaluated [26]. Given its advantages, bifactor modeling has been recommended and applied increasingly in health-related research examining the structure of complex constructs that are characterized by a strong general factor yet at the same time show evidence of multidimensionality [27,28]. In this study, the bifactor model was also applied as an alternative structure of the DERS, not only to solve the inconsistencies in previous studies, but also to provide researchers with a new perspective and more information to understand the underlying factor structure of the DERS.

## 2. Method

### 2.1. Participants

The participants were adolescents and adults from 32 cities in China. Participants had to sign a written informed consent and complete the questionnaire including demographic data and self-report questionnaires via an online crowdsourcing platform in Chinese mainland. The final sample was 1036 participants and the range of their age was 12–66 years (*M*age = 31.21 years, *SD*age = 13.98). The sample included 636 females (61.4%) and 400 males (38.6%). There were no significant sex differences on the age with *t* (758) = −1.186 and with *p* = 0.236. In terms of region, they came from the rural and urban areas in China by 45.3% and 54.7% respectively. This study was approved by the local Ethics Committee of School of Psychology, South China Normal University (code number: SCNU-PSY-2019-3-70).

### 2.2. Measures

#### 2.2.1. DERS (Chinese Version)

The Difficulties in Emotion Regulation Scale (DERS) was developed by Gratz and Roemer [8] and was translated into Chinese by Li et al. [16]. The Chinese version of DERS is a 36-item self-reporting questionnaire with a 5-point Likert scale ranging from 1 (almost never) to 5 (almost always). Higher total score indicates greater difficulties in ER. In this study, the Chinese version of DERS has a Cronbach’s alpha of 0.89 and a split-half reliability of 0.87.

#### 2.2.2. Reevaluation of Life Orientation Test (Chinese Version)

The Life Orientation Test (LOT) [29] and its revision, the Reevaluation of Life Orientation Test (LOT-R) [30], were widely used to assess dispositional optimism. In this study, the Chinese version of LOT-R translated by Li [31] is chosen as the calibration scale. It contains 10 items, including 4 filling items, 3 positive items, and 3 negative items. The items were answered on a 5-point Likert-type scale ranging from 0 (strongly disagree) to 4 (strongly agree). The total score is the general index of optimism tendency, while the positive and negative item scores were used to measure optimism and pessimism, respectively. In this study, the Chinese version of LOT-R has a Cronbach’s alpha of 0.89 and a split-half reliability of 0.82.

### 2.3. The Procedure of Analyses

In this study, third set of analysis were conducted to investigate the structure of the DERS. SPSS version 23 [32] and R package “psych” [33] were applied in exploratory factor analysis (EFA), and Mplus7.0 [34] was used for confirmatory factor analysis (CFA).

In the first set of analysis, CFA was conducted with data from the full sample (*n* = 1036), to confirm the representative competing structures of DERS, as well as to assess whether these existing structures represent the structure of ER best. These structures are introduced in section Factor Structure of the DERS.

In the second set of analysis, all the participants were randomly divided into two groups. One group (*n* = 518) was subjected to the EFA to explore whether there was a new structure fitted the Chinese participants while the other (*n* = 518) was subjected to the CFA to confirm this new structure’s performance. Before implementing EFA, the Kayser–Meyer–Olkin (KMO) and Bartlett’s Test of Sphericity was used to evaluate whether factor analysis is feasible for the current data. Generally, the values of KMO are >0.9 for a good fit, between 0.5 and 0.6 for an acceptable fit, and <0.5 for an unacceptable fit while a significant result of the Bartlett’s Test of Sphericity indicates that the data is suitable for the EFA [35].

In EFAs, determining the number of factors is the major issue and many factor retention methods have been developed. Kaiser’s criterion and the Scree Test [36] are the two mainstream methods since they are easy to implement using software such as SPSS and SAS. The parallel analysis (PA) [37] can offer precise results by comparing the eigenvalues from the actual data with the eigenvalues from the random data, and factors are retained when the eigenvalue from the actual data is higher [38]. PA is one of the most precise methods. However, it is risky to make a decision relying solely on this method, because of its slight tendency to retain too many factors [39]. To make decision with caution, both the conventional methods (i.e., Kaiser’s criterion and the Scree Test) and more robust empirical criteria (i.e., PA) were used in the current study.

In the third set of analysis, bifactor CFA was performed on structures mentioned in section Factor Structure of the DERS, and the results of the bifactor CFA will be compared with the traditional model-fit. After a series of comparisons, the best structure representative was determined. To evaluate the importance of the general factor accounting for item variance, we examined the proportion of variance in scale scores, denoted by omega hierarchical (*ω_h_*). The value of *ω_h_* ranges from 0 to 1, and a higher value means stronger influence for scale scores from the general factor shared by all items [40]. Besides, proportion of explained common variance (ECV) as a useful index was calculated to determine the importance of general factor. The cut-off value of ECV in a bifactor model is 60% and a higher value means a better performance [40].

### 2.4. The Goodness-of-Fit Indices

Note that all of the above-mentioned CFA procedures were conducted using five evaluation criteria, including chi-square to degrees of freedom (*χ*^2^*/df*), root mean square error of approximation (RMSEA), standardized root-mean-square residual (SRMR), comparative fit index (CFI), and Tucker-Lewis Index (TLI). The standards for these evaluation criteria of CFA [41] were shown in Table 1.

Additionally, the chi-square difference test was also employed to compare between structures. The lower the value of chi-square is, the better the fitness of the structure is. It is necessary to note that the chi-square test is sensitive to the size of the sample, and it may be significant when the actual difference between the observed model and the implied model covariance is small.

## 3. Results

### 3.1. Descriptive Characteristics

Descriptive statistics of the DERS items in the sample of 1036 Chinese adolescents and adults were presented in the Table 2. The average score of the DERS was 116.31 (*SD* = 18.61), of which the mean value of males was 118.98 (*SD* = 18.77) and that of females was 114.63 (*SD* = 18.31). The total skewness was 0.22, indicating that the sample was slightly skewed to the right; the total kurtosis was −0.59, meaning that the sample’s peak was slightly flat. Overall, the measured data of each variable conformed to a normal distribution.

### 3.2. Testing Existing Structures

Several representative original structures of the DERS were selected, and CFA was conducted on them. Details of the selected structures of the DERS can be seen in section Factor Structure of the DERS. As shown in Table 3, each existing structure reached the close fitting on the SRMR. Besides, model A and model C were not the acceptable fitting on the RMSEA. In addition, all models were much larger than 5 on the *χ^2^/df*. Except for model A, the remaining models were not acceptable on the CFI and TLI. Taken together the *χ*^2^/*df*, RMSEA, SRMR, CFI, and TLI, none of the structures met the acceptable fitting levels. In order to obtain an acceptable model for the collected data, the next step was to explore and verify the new factor structures.

### 3.3. EFA and CFA for Alternative Structure

From the CFA, it was assumed that a new multidimensional structure might exist. Thus, the data was divided into two halves randomly and subjected to the EFA and CFA, respectively. The KMO’s Test of Sampling Adequacy was 0.83 and the Bartlett’s Test of Sphericity (*χ*^2^ = 7499.39, *df* = 630) was significant (*p* < 0.001), indicating that the DERS was appropriate for a factor analysis.

Principal axis factoring with oblique rotation as the method of factor extraction was used in this study and the cut-off value of the factor loadings was 0.40 (i.e., 16% of the common variance). Three retention methods, as mentioned before, were implemented to determine the number of factors. According to the result of PA, nine factors were retained and the results of the factor rotation were shown in the left panel in Table 4. This 9-factor solution could account for 49% of the total variance. Based on the result of the Kaiser criterion and the Scree Test [36], ten factors were retained and the results of the factor rotation were shown in the right panel in Table 4. This 10-factor solution could account for 51.31% of the total variance. In these two factor solutions, all items succeeded to load moderately to strongly (0.31–0.93) on the factors, with the communalities (*h*^2^) of these items ranged from 0.22 to 1.01. Each factor was defined by three or more items.

To verify the newly factor structures, the remaining half data was submitted to CFA. However, none of new structures provided acceptable fit to the data in this sample, comprehensively based on all goodness-of-fit indexes. In consequence, both new structures from EFA were not considered to the bifactor CFA.

### 3.4. Bifactor CFA

For the bifactor CFA, we extracted the general factor of ER to form the bifactor model based on the models in Table 3. Results of bifactor CFA fitting for these models are shown in Table 5. Obviously, the goodness-of-fit of the bifactor models was much higher than that of the original models. Except that the TLI of the six-specific-factor bifactor model [15] was slightly lower than 0.90, all indices of each model were within the acceptable or close range. Compared with CFA results, the four-specific-factor bifactor model of DERS had the best goodness-of-fit, whose RMSEA was about 0.05, SRMR was less than 0.08, CFI was more than 0.95, and TLI was about 0.95. Besides, the DIFFTEST [the bifactor model based on model A vs. the bifactor model based on model C: Δ*χ*^2^ = 394.63 **, Δ*df* = 22] indicated that the four-specific-factor bifactor model provided a significantly better fit to the DERS (selected models are marked with “#” in Table 5).

The item-factor loadings from the standardized bifactor CFA solution of the best fit model are shown in Figure 2. Regarding the general factor, all item-factor loadings were greater than 0.70, demonstrating that all items of DERS had a strong impact on the general factor. All CFA loading estimates were statistically significant (*p* < 0.05). In the bifactor model, the *ω_h_* of the general factor was 81% and the ECV of the general factor was 89%. This demonstrated that the general ER factor of bifactor model for the DERS accounted for 81% of the variance of the summed score and 89% of the common variance of all items, which suggested that that only a small part of the variance in the subscale scores could be explained by the specific factors, beyond what was already accounted for by the general factor. This further demonstrated that the DERS might be best represented as a unidimensional construct and the items of DERS had stronger associations with the general ER factor than with the specific factors.

### 3.5. Criterion-Related Validity

To test the relationship among variables and the criterion validity, bivariate correlation analysis was conducted for the general and specific factors (see Table 6). Both the total scale and the four subscale scores showed good internal reliability in all samples (full sample, male, and female), with Cronbach’s *α* ranging from 0.62 [0.53,0.71] to 0.89 [0.87,0.90]. The scores on the four subscales and total scale of the DERS were strongly inter-correlated (*r* ranged from 0.39 to 0.92) in all three samples. With LOT-R conducted as an external validation criterion, the result of correlation analysis showed that the total scale and four subscales significantly and positively associated (*r* ranged from 0.67 to 0.92) with the criterion. Besides, the coefficients of each indicator in subscale1 ranged from 0.54 to 0.90, subscale2 ranged from 0.55 to 0.83, subscale3 ranged from 0.68 to 0.90, and subscale4 ranged from 0.65 to 0.81.

To test the influence of sex on the scores of each subscale, the one-way MANOVA was conducted. The results showed that there was a significant multivariate main effect for sex (*F*_(4,1034)_ = 5.51, *p* < 0.001, *η*^2^ = 0.02). In order to control Type Ⅰ error rate across the four following univariate ANOVAs, we used the Bonferroni method at a *α* = 0.01 level of significance. On the four subscales, scores of male participants were significantly higher than female participants’, with *F*_(1,1034)_ = 10.43, *p* < 0.001, *η*^2^ = 0.01 on subscale1, *F*_(1,1034)_ = 14.42, *p* < 0.001, *η*^2^ = 0.01 on subscale2, *F*_(1,1034)_ = 17.23, *p* < 0.001, *η*^2^ = 0.02 on subscale3, *F*_(1,1034)_ = 3.75, *p* < 0.05, *η*^2^ = 0.004 on subscale4.

## 4. Discussion

Although considerable studies have already been conducted to investigate the structure of the DERS, little is known about its underlying factor structure. This study was implemented in the sample of 1036 Chinese adolescents and adults, aiming at revealing the most optimal structure and the latent factor relations underlying the DERS measure via bifactor modeling. After a series of comparisons, we concluded that the bifactor model offered a better fit to the current data than the correlated traits model. The fit indices proved that among several alternative models, the four-specific-factor bifactor structure of the DERS was the most optimal. Furthermore, the structural analysis based on the optimal bifactor model brought a new perspective to conceptualize the structure of ER from both general and specific factors.

In addition to fit indices, some statistical indices associated with the bifactor model were also examined in this study. For example, the *ω_h_* and ECV statistic could evaluate the importance of the general factor. In the bifactor model of this study, the *ω_h_* of the general factor was 81% (above the cut-off of 70%) and the ECV of the general factor was 89% (above the cut-off of 60%). These findings suggested this study should continue to focus on a single core construct and support the existence of a dominant global ER dimension, which explained over two thirds of common variance in the DERS item responses and correlated significantly and moderately with the criterion measure of dispositional optimism. It is worth noting that although a considerable amount of variance was accounted for by the general factor (i.e., ER), the traditional single factor structure is not recommended to fit the current sample data. One of the primary reasons is that a number of studies indicated that the DERS is a multidimensional measure. Additionally, the single-factor structure of the DERS presented a poor fit (RMSEA = 0.07, SRMR = 0.04, CFI = 0.86, and TLI = 0.85). Therefore, compared with the single-factor model, the bifactor model can better describe and generalize the structure of DERS. Together with previous reports of a strong general factor in several adult samples [42], this finding lends further confidence to the common practice of using the total DERS score as a general index of ER [43].

A notable strength and novel contribution of this study, using bifactor analysis, is that the effectiveness of items could be assessed in two aspects (i.e., the loadings on the general factor and the specific factor), which is difficult in traditional correlated factors model analysis. A correlated factors model did not include a general factor and attributes all explanatory variance to first-order factors [44]. A correlated factors model is conceptually ambiguous because it is not able to separate the specific or unique contributions of a factor from the effect of the overall construct shared by all interrelated factors [45], whereas a bifactor model contains a general factor (G) and multiple specific factors (S). Because G and S are independent, a Bifactor model can disentangle how each factor contributes to the systematic variance in each item. The possibility of segmenting the variance in independent sources is one of the primary advantages of the Bifactor model. An additional benefit of bifactor modeling is that the relation between domain-specific factors and criterion variables can be examined while holding the general factor constant [46]. This approach might provide evidence of the incremental utility of domain-specific factors, beyond the general factor, in predicting psychological constructs theoretically relevant to emotion dysregulation. More specifically, through the use of bifactor modeling, we can determine whether the domain-specific factors of the DERS account for unique variance in relevant criterion variables after accounting for the DERS general factor. This study also implied that the use of multidimensional latent variable model specifications that account for the general and domain specific factors (e.g., bifactor models) should be considered when examining the structures of related ER measures.

The present study benefited from a relatively large sample and the implementation of sophisticated modern measurement methods to specifically reveal the underlying factor structure of DERS. With the current results in mind, we recommend that researchers use an investigative approach, such as a bifactor analysis, to inspect the extent to which the variance in item responses is due to a general or specific factor when assessing the structure of psychological measures (with data that indicates construct-relevant multidimensionality). In sum, IRT-based structural studies are suitable for informing the development of ER construct and the measures that emerge for assessing ER. As such, continued structural work of this type is needed to inform our understanding of common and domain specific components of the ER.

The use of a bifactor-analytic procedure provided us with the opportunity to assess the unique contributions of each item to a general and a domain-specific factor. Although we think that bifactor modeling is an interesting psychometric tool to investigate the data structure in psychological measures, there are also some issues that need further attention. First, only individuals from the general community were included. It would be important to replicate the analyses performed in the current study in clinical samples, in order to ascertain whether the DERS could potentially explain systematic individual differences in emotion regulation skills among clinical populations. Second, because of the self-report method we utilized, it was not possible to establish a standardized environment for data collection. Thus, uncontrollable environmental factors in the data collection process may have influenced the results. Third, we relied solely on self-reported data in all the analyses; thus, the findings could be accounted for by method variance [47]. Future studies should receive data from multiple sources, involving direct observation and semistructured interviews, when examining the properties of the DERS or its revision.

## 5. Conclusions

Despite these limitations, the bifactor method has been successfully used to resolve similar inconsistencies in the factor structure of the DERS in this study, producing better-fitting model in a sample of adolescents and adults. This article brought a new perspective and more information to understand the underlying factor structure of the DERS.

## Figures and Tables

**Figure 1 ijerph-18-04208-f001:**
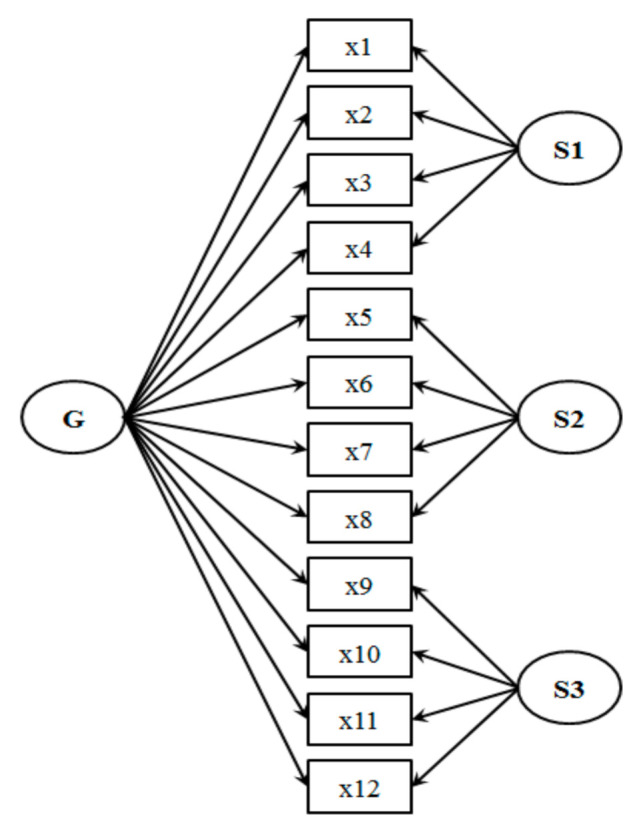
A bifactor model with three specific factors.

**Figure 2 ijerph-18-04208-f002:**
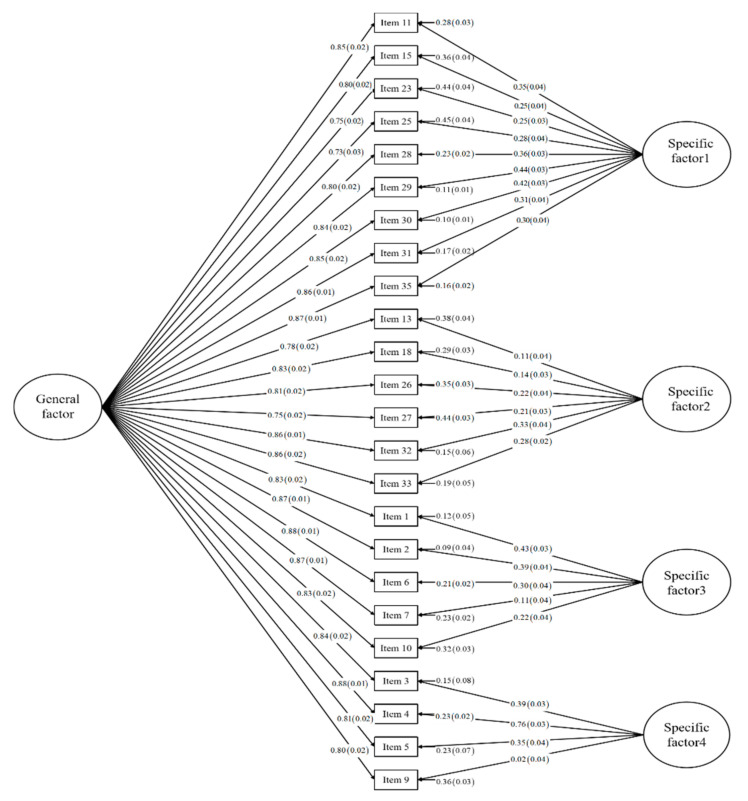
Bifactor structure of the DERS: Item-factor loadings from the ideal CFA solution (*N* = 1036). Note: Specific factor1 = Non-acceptance; Specific factor2 = Goals; Specific factor3 = Awareness; Specific factor4 = Clarity.

**Table 1 ijerph-18-04208-t001:** The standard for five evaluation criteria of CFA.

Standard	*χ*^2^/*df*	RMSEA	SRMR	CFI	TLI
an acceptable fit	<5	<0.08	≤0.10	≥0.90	≥0.90
a close fit	<3	≤0.05	≤0.08	≥0.95	≥0.95

Note: *df* = degrees of freedom; RMSEA = root mean square error of approximation; SRMR = standardized root-mean-square residual; CFI = comparative fit index; TLI = Tucker-Lewis Index.

**Table 2 ijerph-18-04208-t002:** Descriptive statistics of the DERS items (*N* = 1036).

Item	*M*(*SD*) Total	*M*(*SD*) Male	*M*(*SD*) Female	Skewness	Kurtosis
Y1	3.35 (1.04)	3.50 (1.03)	3.25 (1.04)	−0.29	−0.69
Y2	3.36 (1.14)	3.53 (1.12)	3.24 (1.13)	−0.34	−0.84
Y3	2.86 (1.00)	2.87 (0.96)	2.85 (1.02)	−0.03	−0.62
Y4	3.83 (0.85)	3.89 (0.86)	3.79 (0.85)	−0.59	0.33
Y5	3.50 (1.20)	3.52 (1.17)	3.49 (1.22)	−0.28	−0.96
Y6	3.38 (1.05)	3.51 (1.04)	3.30 (1.04)	−0.41	−0.50
Y7	3.46 (0.99)	3.57 (0.98)	3.39 (0.99)	−0.64	0.05
Y8	3.56 (0.93)	3.63 (0.91)	3.51 (0.94)	−0.36	−0.43
Y9	2.88 (0.95)	3.00 (1.00)	2.81 (0.91)	−0.13	−0.63
Y10	3.50 (0.91)	3.56 (0.92)	3.46 (0.91)	−0.36	−0.20
Y11	2.62 (0.91)	2.66 (0.93)	2.60 (0.90)	0.11	−0.28
Y12	3.36 (1.10)	3.45 (1.14)	3.30 (1.07)	−0.33	−0.38
Y13	3.47 (1.03)	3.56 (1.04)	3.41 (1.02)	−0.53	−0.25
Y14	3.02 (0.99)	3.06 (0.96)	3.00 (1.00)	−0.01	−0.39
Y15	3.22 (0.92)	3.29 (0.94)	3.17 (0.90)	−0.26	0.31
Y16	3.10 (1.16)	3.19 (1.15)	3.04 (1.16)	−0.10	−0.68
Y17	3.04 (1.04)	3.02 (0.98)	3.05 (1.08)	0.10	−0.51
Y18	3.29 (1.01)	3.30 (0.96)	3.28 (1.05)	−0.05	−0.45
Y19	3.22 (1.01)	3.23 (0.98)	3.22 (1.04)	−0.02	−0.43
Y20	2.96 (0.85)	3.01 (0.88)	2.93 (0.84)	−0.04	−0.32
Y21	3.25 (0.89)	3.28 (0.99)	3.22 (0.82)	−0.07	−0.40
Y22	3.05 (1.05)	3.10 (1.03)	3.01 (1.07)	−0.13	−0.47
Y23	2.85 (0.88)	2.88 (0.94)	2.83 (0.83)	0.09	−0.02
Y24	3.15 (0.99)	3.11 (0.93)	3.17 (1.02)	0.02	−0.48
Y25	3.13 (0.93)	3.17 (0.96)	3.10 (0.91)	0.01	−0.44
Y26	3.21 (0.99)	3.24 (0.98)	3.19 (1.00)	0.08	−0.35
Y27	3.29 (1.08)	3.40 (1.05)	3.23 (1.10)	0.01	−0.73
Y28	3.35 (1.22)	3.60 (1.20)	3.19 (1.20)	−0.45	−0.69
Y29	3.30 (1.26)	3.50 (1.25)	3.17 (1.25)	−0.39	−0.85
Y30	3.31 (1.16)	3.42 (1.22)	3.24 (1.12)	−0.27	−0.74
Y31	3.46 (1.15)	3.54 (1.08)	3.42 (1.19)	−0.59	−0.35
Y32	3.42 (0.89)	3.45 (0.90)	3.39 (0.88)	−0.27	−0.03
Y33	3.07 (0.93)	3.07 (0.91)	3.06 (0.95)	−0.08	−0.20
Y34	3.21 (0.89)	3.22 (0.92)	3.21 (0.87)	0.00	−0.38
Y35	3.10 (1.22)	3.27 (1.28)	2.99 (1.16)	0.18	−0.86
Y36	3.21 (1.20)	3.38 (1.25)	3.10 (1.16)	0.05	−0.93
Total	116.31 (18.62)	118.98 (18.80)	114.63 (18.32)	0.22	−0.59

**Table 3 ijerph-18-04208-t003:** CFA model-fit results of the existing structures of the DERS (*N* = 1036).

Model	No. of Factors	No. of Items	$\chi^{2}$	*df*	*χ*^2^/*df*	RMSEA [90% CI]	SRMR	CFI	TLI
A (Marin Tejeda et al., 2012) [13]	4	24	2827.43	246	11.49	0.10 [0.09, 0.11]	0.04	0.91	0.90
B (Bardeen et al., 2012) [17]	5	30	4476.45	395	11.33	0.10 [0.09, 0.11]	0.05	0.89	0.88
C (Guzmán-González et al., 2014) [12]	5	25	3585.68	265	13.53	0.11 [0.10, 0.12]	0.03	0.89	0.88
D (Gómez-Simón et al., 2014) [14]	6	36	6383.28	579	11.02	0.09 [0.08, 0.10]	0.05	0.88	0.87
E (Gratz & Roemer, 2004) [8]	6	36	6213.31	579	10.73	0.10 [0.09, 0.11]	0.05	0.88	0.87
F (Li et al., 2018) [15]	6	32	5095.30	449	11.35	0.10 [0.09, 0.11]	0.04	0.88	0.87

Note: *df* = degrees of freedom; 90% CI = 90% confidence interval for RMSEA.

**Table 4 ijerph-18-04208-t004:** Item-factor loadings from EFA for the nine-factor solution and ten-factor solution.

Items	9-Factor Solution									*h* ^2^	10-Factor Solution										h^2^
	1	2	3	4	5	6	7	8	9		1	2	3	4	5	6	7	8	9	10	
Y1	**0.51**	0.09	0.02	0.08	0.07	−0.06	−0.03	0.04	0.11	0.30	0.38	0.08	0.01	0.06	0.03	0.00	0.02	0.06	0.05	**0.63**	0.56
Y2	**0.66**	0.03	0.10	0.16	0.27	0.06	−0.05	0.01	0.15	0.57	0.53	0.01	0.10	0.14	0.26	0.14	−0.01	0.11	0.01	**0.60**	0.77
Y3	−0.03	0.21	0.27	0.03	0.01	0.00	**0.79**	0.03	0.06	0.75	−0.02	0.22	0.27	0.02	0.02	−0.01	**0.76**	0.06	0.05	−0.04	0.71
Y4	**0.34**	0.18	0.19	0.13	0.09	0.21	0.16	−0.04	0.02	0.28	**0.36**	0.19	0.20	0.14	0.12	0.18	0.14	0.03	−0.05	−0.03	0.30
Y5	0.10	0.11	0.09	−0.01	0.09	0.03	**0.64**	0.04	0.02	0.45	0.09	0.11	0.08	−0.02	0.09	0.04	**0.67**	0.02	0.04	0.04	0.49
Y6	**0.73**	−0.05	0.05	0.04	0.11	0.09	0.04	0.09	0.02	0.57	**0.70**	−0.04	0.06	0.04	0.12	0.09	0.03	0.03	0.09	0.17	0.56
Y7	**0.61**	0.03	0.04	0.15	0.15	0.24	0.09	0.04	0.05	0.49	**0.62**	0.03	0.05	0.16	0.18	0.21	0.08	0.07	0.03	0.05	0.50
Y8	**0.76**	−0.02	−0.03	−0.03	0.05	0.02	0.04	0.01	0.03	0.59	**0.78**	−0.01	−0.02	−0.01	0.08	−0.03	0.02	0.06	0.00	0.06	0.62
Y9	0.08	0.13	**0.41**	0.09	0.15	0.03	0.19	0.17	0.07	0.29	0.10	0.14	**0.41**	0.10	0.16	0.01	0.18	0.07	0.18	−0.07	0.31
Y10	**0.55**	0.04	0.00	0.00	−0.02	0.13	0.00	0.02	−0.03	0.32	**0.59**	0.06	0.00	0.01	0.01	0.08	−0.03	−0.01	0.00	−0.01	0.36
Y11	−0.06	0.11	**0.52**	0.09	−0.04	−0.06	0.11	0.11	0.15	0.35	−0.06	0.11	**0.51**	0.09	−0.04	−0.06	0.11	0.14	0.13	0.01	0.34
Y12	0.25	0.09	0.23	0.14	**0.39**	0.21	0.00	−0.06	0.05	0.35	0.25	0.09	0.24	0.14	**0.31**	0.19	−0.01	0.06	−0.07	0.02	0.29
Y13	**0.44**	−0.02	0.06	−0.03	0.08	−0.06	0.00	−0.08	−0.03	0.22	**0.43**	−0.03	0.07	−0.02	0.10	−0.08	−0.01	−0.02	−0.07	0.06	0.22
Y14	0.05	**0.93**	0.19	0.06	0.04	0.12	0.09	0.17	0.03	0.96	0.04	**0.93**	0.29	0.05	0.04	0.12	0.09	0.03	0.17	0.04	1.01
Y15	0.11	0.13	0.15	0.13	0.11	0.13	0.02	0.00	**0.49**	0.34	0.08	0.13	0.15	0.13	0.11	0.15	0.02	**0.46**	0.01	0.10	0.32
Y16	0.07	0.06	0.11	0.00	0.04	0.04	0.05	0.15	**0.77**	0.64	0.06	0.07	0.11	0.00	0.05	0.03	0.04	**0.83**	0.14	0.00	0.73
Y17	0.04	**0.50**	0.18	−0.08	0.06	−0.09	0.10	0.04	0.26	0.38	0.03	**0.51**	0.18	−0.07	0.07	−0.11	0.09	0.26	0.03	0.02	0.39
Y18	−0.04	**0.62**	0.19	0.05	0.03	0.14	0.20	0.19	−0.01	0.52	−0.05	**0.62**	0.19	0.04	0.02	0.14	0.21	−0.01	0.19	0.02	0.53
Y19	0.04	**0.75**	0.21	0.09	0.08	0.09	0.04	0.08	0.05	0.64	0.04	**0.74**	0.21	0.09	0.08	0.09	0.04	0.05	0.08	0.03	0.63
Y20	0.11	0.22	0.02	−0.03	0.03	0.02	0.01	**0.45**	0.10	0.28	0.11	0.23	0.01	−0.03	0.04	0.02	0.01	0.11	**0.44**	0.01	0.27
Y21	0.10	0.19	**0.67**	0.02	−0.01	0.15	0.01	−0.09	0.04	0.53	0.10	0.19	**0.68**	0.02	0.00	0.13	0.01	0.05	−0.09	0.01	0.54
Y22	0.18	0.13	0.01	0.07	−0.07	0.11	0.15	**0.32**	0.23	0.25	**0.31**	0.15	0.00	0.07	−0.05	0.29	0.14	0.24	0.29	−0.02	0.37
Y23	−0.10	0.14	**0.54**	0.08	0.06	−0.02	0.15	0.15	0.12	0.39	−0.13	0.14	**0.53**	0.07	0.05	0.01	0.16	0.11	0.17	0.06	0.40
Y24	0.23	0.20	−0.02	0.04	0.11	0.00	0.06	0.21	**0.35**	0.28	**0.35**	0.22	−0.03	0.04	0.12	0.23	0.05	0.02	0.19	−0.01	0.28
Y25	0.16	0.18	**0.70**	−0.01	0.04	0.09	0.01	−0.05	−0.04	0.56	0.15	0.18	**0.70**	−0.01	0.05	0.08	0.01	−0.04	−0.04	0.03	0.56
Y26	0.18	−0.08	−0.28	0.05	0.07	**0.49**	−0.09	−0.40	0.07	0.54	0.18	−0.08	−0.27	0.04	0.07	**0.48**	−0.09	0.07	−0.43	0.03	0.55
Y27	0.13	0.18	0.21	0.04	0.22	**0.63**	0.02	0.05	0.08	0.55	0.14	0.19	0.21	0.03	0.22	**0.64**	0.02	0.07	0.04	0.00	0.57
Y28	0.06	0.07	0.11	**0.43**	0.15	0.17	0.07	0.08	0.10	0.28	0.03	0.07	0.11	**0.42**	0.14	0.22	0.09	0.08	0.08	0.09	0.29
Y29	0.11	0.03	0.09	**0.84**	0.15	0.09	0.02	−0.02	0.00	0.76	0.10	0.03	0.09	**0.84**	0.16	0.10	0.02	0.00	−0.02	0.01	0.76
Y30	0.12	0.02	0.05	**0.93**	0.14	−0.02	−0.03	0.00	0.08	0.91	0.09	0.02	0.05	**0.93**	0.16	0.00	−0.04	0.08	0.01	0.06	0.91
Y31	0.17	0.06	0.18	0.15	0.19	**0.32**	−0.01	−0.02	0.08	0.23	0.16	0.06	0.18	0.24	0.29	**0.33**	−0.01	0.07	−0.02	0.06	0.32
Y32	0.06	0.14	0.17	0.01	0.08	0.20	**0.45**	0.26	0.09	0.38	0.06	**0.45**	0.24	0.01	0.09	0.20	0.17	0.09	0.26	−0.01	0.42
Y33	0.01	0.19	0.23	0.05	0.12	−0.05	0.05	**0.56**	0.07	0.43	−0.02	0.19	0.21	0.04	0.11	0.00	0.06	0.06	**0.59**	0.10	0.46
Y34	**0.48**	0.00	−0.05	0.04	−0.02	0.10	0.01	0.10	0.03	0.26	**0.48**	0.01	−0.04	0.04	−0.01	0.08	0.00	0.04	0.08	0.07	0.25
Y35	0.20	0.06	0.00	0.18	**0.87**	0.15	0.10	0.09	0.06	0.88	0.17	0.07	0.00	0.17	**0.86**	0.16	0.10	0.06	0.09	0.08	0.86
Y36	0.24	0.10	0.06	0.19	**0.77**	0.13	0.08	0.05	0.09	0.73	0.22	0.10	0.06	0.19	**0.79**	0.12	0.07	0.09	0.05	0.05	0.75

Note: *h*^2^ = communality; The bold values indicate salient factor loadings.

**Table 5 ijerph-18-04208-t005:** Bifactor CFA model-fit results of the suggested structures of the DERS (*N* = 1036).

Model	No. of Factors	No. of Items	$\chi^{2}$	*df*	*χ*^2^/*df*	RMSEA [90% CI]	SRMR	CFI	TLI
Bi-A (based on Marin Tejeda et al. 2012) [13] ^#^	4	24	606.35 *	228	2.66	0.04 [0.03, 0.05]	0.02	0.97	0.96
Bi-B (based on Bardeen et al. 2012) [17]	5	30	1624.76 *	375	4.33	0.05 [0.04, 0.06]	0.03	0.93	0.92
Bi-C (based on Guzmán-González et al. 2014) [12]	5	25	1000.98 *	250	4.00	0.05 [0.04, 0.06]	0.03	0.94	0.93
Bi-D (based on Gómez-Simón et al. 2014) [14]	6	36	2299.44 *	558	4.12	0.05 [0.04, 0.06]	0.03	0.92	0.91
Bi-E (based on Gratz and Roemer 2004) [8]	6	36	2408.538 *	558	4.32	0.05 [0.04, 0.06]	0.03	0.91	0.90
Bi-F (based on Li et al. 2018) [15]	6	32	1996.571 *	432	4.62	0.05 [0.04, 0.06]	0.03	0.91	0.90

Note: Bi- referred to the bifactor model based on the relevant existing model; ^#^ the best fit model; * *p* < 0.05.

**Table 6 ijerph-18-04208-t006:** Correlations between factors of the DERS and LOT-R according to the bifactor structure.

	Mean	SD	α [90%CI]	Subscale1	Subscale2	Subscale3	Subscale4	LOT-R
Full sample (*N* = 1036)								
Total scale	54.19	13.01	0.89 [0.87,0.90]	0.92 **	0.73 **	0.80 **	0.72 **	0.92 **
Subscale1	19.33	6.24	0.82 [0.80,0.84]	–	0.56 **	0.63 **	0.56 **	0.90 **
Subscale2	13.74	3.04	0.63 [0.58,0.68]	–	–	0.43 **	0.45 **	0.72 **
Subscale3	12.04	3.90	0.82 [0.80,0.84]	–	–	–	0.44 **	0.79 **
Subscale4	9.07	2.75	0.72 [0.68,0.76]	–	–	–	–	0.76 **
Male (*N* = 400)								
Total scale	56.3	12.97	0.88 [0.86,0.90]	0.92 **	0.69 **	0.80 **	0.70 **	0.91 **
Subscale1	20.32	6.44	0.83 [0.80,0.86]	–	0.52 **	0.63 **	0.54 **	0.90 **
Subscale2	14.03	2.97	0.62 [0.53,0.71]	–	–	0.40 **	0.39 **	0.67 **
Subscale3	12.67	3.92	0.83 [0.80,0.86]	–	–	–	0.45 **	0.79 **
Subscale4	9.28	2.69	0.69 [0.62,0.76]	–	–	–	–	0.74 **
Female (*N* = 636)								
Total scale	52.86	12.88	0.88 [0.87,0.89]	0.91 **	0.75 **	0.78 **	0.72 **	0.92 **
Subscale1	18.71	6.03	0.81 [0.79,0.83]	–	0.58 **	0.62 **	0.57 **	0.90 **
Subscale2	13.57	3.08	0.63 [0.56,0.70]	–	–	0.44 **	0.47 **	0.73 **
Subscale3	11.64	3.84	0.80 [0.78,0.82]	–	–	–	0.42 **	0.78 **
Subscale4	8.94	2.78	0.63 [0.58,0.68]	–	–	–	–	0.76 **

Note: Subscale1 = Non-acceptance; Subscale2 = Goals; Subscale3 = Awareness; Subscale4 = Clarity; ** *p* < 0.01.

## Data Availability

The data presented in this study are available on request from the first author.

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
