# Peer review of "Examining the Structure of Difficulties in Emotion Regulation Scale with Chinese Population: A Bifactor Approach"

_ijerph, 2021, doi:10.3390/ijerph18084208_

Round 1

Reviewer 1 Report

The authors should emphasize the benefit of introducing a general factor and the 4 subfactors, point out the fit criteria of the bifactor model, beyond the replication of the model of Tejeda et al. Also, the association coefficients of each indicator with their respective associated error values. Perhaps a graphical representation of the model could help the reader to better understand the proposal of the study.  

The authors should explain why the items have a strong association with the general factor and less with the different subfactors. Doesn't this result suggest that the test is unidimensional?

Author Response

Response to Reviewer 1 Comments

Point 1: The authors should emphasize the benefit of introducing a general factor and the 4 subfactors, point out the fit criteria of the bifactor model, beyond the replication of the model of Tejeda et al. Also, the association coefficients of each indicator with their respective associated error values. Perhaps a graphical representation of the model could help the reader to better understand the proposal of the study.

Response 1: Thanks for your constructive comments. In the Discussion section of the revised manuscript, we have emphasized the benefit of introducing the four-specific-factor bifactor structure of the DERS (see page 11 lines 345-362 highlighted in red). The specific description can also be seen in the following paragraphs.

This study was implemented in the sample of 1036 Chinese adolescents and adults, aiming at revealing the most optimal structure and the latent factor relations underlying the DERS measure via bifactor modeling. After a series of comparisons, we concluded that among several alternative models, the four-specific-factor bifactor structure of the DERS was the most optimal (RMSEA = 0.04, SRMR = 0.02, CFI = 0.97, and TLI= 0.96). Compared with the correlated traits model, there are some benefits of introducing the four-specific-factor bifactor structure of the DERS.

On the one hand, a notable strength and novel contribution of this study, using bifactor analysis, is that the effectiveness of items could be assessed in two aspects (i.e., the loadings on the general factor and the specific factor), which is difficult in traditional correlated factors model analysis. A correlated factors model did not include a general factor and attributes all explanatory variance to first-order factors [44]. A correlated factors model is conceptually ambiguous because it is not able to separate the specific or unique contributions of a factor from the effect of the overall construct shared by all interrelated factors [45], whereas a bifactor model contains a general factor (G) and multiple specific factors (S). Because G and S are independent, a Bifactor model can disentangle how each factor contributes to the systematic variance in each item. The possibility of segmenting the variance in independent sources is one of the primary advantages of the Bifactor model.

On the other hand, an additional benefit of bifactor modeling is that the relation between domain-specific factors and criterion variables can be examined while holding the general factor constant [46]. This approach might provide evidence of the incremental utility of domain-specific factors, beyond the general factor, in predicting psychological constructs theoretically relevant to emotion dysregulation. More specifically, through the use of bifactor modeling, we can determine whether the domain-specific factors of the DERS account for unique variance in relevant criterion variables after accounting for the DERS general factor.

In the revised manuscript, the coefficients of each indicator in each subscale had been added in the Results section (see page 10 lines 304-306 highlighted in red). To help the reader to better understand the proposal of the study, standardized factor loadings for the bifactor confirmatory model with four specific factors (i.e., Figure 2) had been added in the Results section (see page 9 lines 292-294 highlighted in red). Note that the measurement error values of each indicator had been presented in Figure 2.

Point 2: The authors should explain why the items have a strong association with the general factor and less with the different subfactors. Doesn't this result suggest that the test is unidimensional?

Response 2: Thanks for your constructive comments. Empirical evidence indicated that the total score of DERS can be considered essentially unidimensional. A Bifactor CFA model analysis of the DERS formulated to test the importance of the general factor yielded the following fit indices: RMSEA = 0.04, SRMR = 0.02, CFI = 0.97, and TLI= 0.96. These indices fit well with each other, indicating an effective model fit. All item-factor loadings in the general factor were greater than 0.70, demonstrating that all items of DERS had a strong impact on the general factor. All CFA loading estimates were statistically significant (p < 0.05). Besides, in the bifactor model, the  of the general factor was 81% (above the cut-off of 70%) and the ECV of the general factor was 89% (above the cut-off of 60%). That meant that only a small part of the variance in the subscale scores could be explained by the specific factors, beyond what was already accounted for by the general factor. This further demonstrated that the DERS might be best represented as a unidimensional construct and the items of DERS had stronger associations with the general ER factor than with the specific factors.

It is worth noting that although a considerable amount of variance was accounted for by the general factor (i.e., ER), the traditional single factor structure is not recommended to fit the current sample data. One of the primary reasons is that a number of studies indicated that the DERS is a multidimensional measure. Additionally, the single-factor structure of the DERS presented a poor fit (RMSEA = 0.07, SRMR = 0.04, CFI = 0.86, and TLI= 0.85). Therefore, compared with the single-factor model, the bifactor model can better describe and generalize the structure of DERS.

In conclusion, the bifactor model demonstrates superior fit to all other simple-structure solutions. This also further explains why the items had stronger associations with the general ER factor than with the specific factors. The explanation had been added in Results section (see page 9 lines 281-291 highlighted in red) and Discussion section (see page 10 lines 328-331 and page 11 lines 332-342 highlighted in red).

Reviewer 2 Report

The authors discuss their findings that result in a bifactor model, with a general ER factor and four distinct sub-dimensions, being the most optimal structure for the Difficulties in Emotion Regulation Scale (DERS).

In lines 125-126, the following sentence needs further clarification, “Unlike the correlated-factors model utilized in past studies, a bifactor structure can easily accommodate the unique nature of the outcomes items by modeling them as part of the general ER dimension shared among all the ER measures items [25]. “

And the following lines 127 -  132 , Moreover, because any remaining systematic covariation among the items would be captured by their loadings on the narrower specific factors, the bifactor approach also makes it possible to estimate the relative sizes of the general and specific symptom components, and to compare their independent contributions to the prediction of external criteria – an important consideration in deciding whether multidimensional assessment is feasible or not [26].”   What is meant by “loadings on the narrower specific factors”, and “compare their independent contributions to the prediction of external criteria”.

Method

The female participants had an age range of 12 - 66 years, or the age range of the entire set of participants were 12 - 66 years.  Additionally, why would there only be an average and standard deviation for male and not female?

Line 156 – grammar - (from 1 almost never to 5 almost always).   Need to re-word.

Line 231 - Several representative original structures of DERS were selected to CFA, with more details in section Factor Structure of the DERS.  Need to re-word – not sure what the authors are saying.  What are the original structures of DERS, how were they selected by CFS, using more details in section Factor Structure of the DERS. 

Line 235 – “At the same time, excepted model A,”  - Should be – “Except for model A,….”

Need to put the title of Table 4 on its own line.

What is the table on page 8 – is it still part of Table 4?

Author Response

Response to Reviewer 2 Comments

The authors discuss their findings that result in a bifactor model, with a general ER factor and four distinct sub-dimensions, being the most optimal structure for the Difficulties in Emotion Regulation Scale (DERS).

Point 1: In lines 125-126, the following sentence needs further clarification, “Unlike the correlated-factors model utilized in past studies, a bifactor structure can easily accommodate the unique nature of the outcomes items by modeling them as part of the general ER dimension shared among all the ER measures items [25].”

Response 1: Thanks for your constructive comments. We are sorry for not describing clearly the characteristics of bifactor model. Morgan et al. [44] suggested that a correlated factors model did not include a general factor and attributes all explanatory variance to first-order factors. A correlated factors model is conceptually ambiguous because it is not able to separate the specific or unique contributions of a factor from the effect of the overall construct shared by all interrelated factors [45]. But bifactor model hypothesizes that (a) there is a general factor that accounts for the commonality shared by the facets, and (b) there are multiple specific factors, each of which accounts for the unique influence of the specific component over and above the general factor.

To help the reader to better understand the meaning of this sentence, we have revised it to “Unlike the correlated-factors model utilized in previous studies, bifactor model hypothesizes that (a) there is a general factor that accounts for the commonality shared by the facets, and (b) there are multiple specific factors, each of which accounts for the unique influence of the specific component over and above the general factor.” (see page 3 lines 125-128 highlighted in red).

Point 2: And the following lines 127 - 132, Moreover, because any remaining systematic covariation among the items would be captured by their loadings on the narrower specific factors, the bifactor approach also makes it possible to estimate the relative sizes of the general and specific symptom components, and to compare their independent contributions to the prediction of external criteria – an important consideration in deciding whether multidimensional assessment is feasible or not [26].” What is meant by “loadings on the narrower specific factors”, and “compare their independent contributions to the prediction of external criteria”.

Response 2: Thanks for your constructive comments. We are sorry for the unclear description of the bifactor model in this study. The “loadings on the narrower specific factors” here has the same meaning as “loadings on the specific factors”. We use “narrower” to describe “specific factors”. Specifically, the general factor is conceptually broader, reflecting the commonality of all items; while group factors tend to be conceptually narrower, which only account for the unique influence of the specific domain.

Brown [46] suggested the bifactor model can be particularly useful in testing whether a subset of the domain specific factors predict external variables, over and above the general factor, as the domain specific factors are directly represented as independent factors. More specifically, the bifactor model allows the evaluation of one or more group dimensions’ unique contribution to prediction after controlling for the general factor. By separating the predictive power of the general and specific factors, bifactor modeling can determine whether the degree of multidimensionality in a given measure is sufficient to support using subscales or not.

To help the reader to better understand the meaning of this sentence, we have revised it to “Moreover, any remaining systematic covariation among the items would be captured by their loadings on the narrower specific factors. Therefore, the bifactor approach also makes it possible to estimate the relative sizes of the general and specific symptom components, and compare their independent contributions to the prediction of ex-ternal criteria. Through this approach, whether the degree of multidimensionality in a given measure is sufficient to support using subscales or not can be evaluated.” (see page 3 lines 128-129 and page 4 lines 130-134 highlighted in red).

Method

Point 3: The female participants had an age range of 12 - 66 years, or the age range of the entire set of participants were 12 - 66 years. Additionally, why would there only be an average and standard deviation for male and not female?

Response 3: Thanks for your helpful comments. We are sorry for the unclear and improper description of the participants’ information. The age range of the entire set of participants were 12 - 66 years. In addition, both the average and standard deviation were for the entire set of participants instead of male or female. The description had been revised in Method section (see page 4 lines 146-149 highlighted in red). The detailed description can also be seen in the next paragraph.

The final sample was 1036 participants and the range of their age was 12 - 66 years (Mage = 31.21 years, SDage = 13.98). The sample included 636 females (61.4%) and 400 males (38.6%). There were no significant sex differences on the age with t (758) = -1.186 and with p = 0.236.

Point 4: Line 156 – grammar - (from 1 almost never to 5 almost always). Need to re-word.

Response 4: Thanks for your helpful comments. We are sorry for the improper description and this description had been corrected to “The Chinese version of DERS is a 36-item self-reporting questionnaire with a 5-point Likert scale ranging from 1 (almost never) to 5 (almost always).” (see page 4 lines 155-157 highlighted in red).

Point 5: Line 231 - Several representative original structures of DERS were selected to CFA, with more details in section Factor Structure of the DERS. Need to re-word – not sure what the authors are saying. What are the original structures of DERS, how were they selected by CFS, using more details in section Factor Structure of the DERS.

Response 5: Thanks for your helpful suggestions. We are sorry for the unclear and improper description of this sentence. A systematic review of the literature indicated that factor structures of DERS were mainly focused on four-, five-, and six- factor solution. Therefore, several representative structures (i.e., models A~F) were selected in this study. Confirmatory factor analysis (CFA) was conducted with data from the full sample to confirm these representative competing structures of the DERS (see Factor Structure of the DERS section) and to assess whether these existing competing structures best represent the structure of DERS. To express the actual meaning, the abovementioned sentence has been corrected to “Several representative original structures of the DERS were selected, and CFA was conducted on them. Details of the selected structures of the DERS can be seen in Factor Structure of the DERS section.” (see page 6 lines 232-234 highlighted in red).

Point 6: Line 235 – “At the same time, excepted model A,” - Should be – “Except for model A,….”

Response 6: Thanks for your constructive comments. We are sorry for the incorrect description and the abovementioned sentence has been corrected to “Except for model A, the remaining models were not acceptable on the CFI and TLI.” (see page 6 lines 236-237 highlighted in red).

Point 7: Need to put the title of Table 4 on its own line.

Response 7: Thanks for your constructive comments. We apologize for our carelessness. The title of Table 4 had been put on its own line (see page 7 line 264 highlighted in red).

Point 8: What is the table on page 8 – is it still part of Table 4?

Response 8: Thank you for your comments. Indeed, the table on page 8 also is part of Table 4. Table 4 displays the item-factor loadings from EFA for the nine -factor solution and ten-factor solution. For space reasons, the factor loadings of items 10-36 are presented on page 8.